# Identifying Barriers to Reducing Portion Size: A Qualitative Focus Group Study of British Men and Women

**DOI:** 10.3390/nu11051054

**Published:** 2019-05-10

**Authors:** Jennifer Ferrar, Danielle Ferriday, Hendrik J. Smit, Duncan C. McCaig, Peter J. Rogers

**Affiliations:** 1Nutrition and Behaviour Unit, School of Psychological Science, University of Bristol, 12a Priory Road, Bristol BS8 1TU, UK; danielle.ferriday@bristol.ac.uk (D.F.); henk.smit@bristol.ac.uk (H.J.S.); D.McCaig@warwick.ac.uk (D.C.M.); peter.rogers@bristol.ac.uk (P.J.R.); 2National Institute for Health Research Bristol Biomedical Research Centre, University Hospitals Bristol NHS Foundation Trust, University of Bristol, Bristol BS2 8AE, UK

**Keywords:** portion reduction, portion size intervention, qualitative, meal satisfaction, focus group

## Abstract

Reducing portion size might reduce meal satisfaction, which could minimize adherence to portion size interventions. The present study sought to identify the perceived barriers for consumers to eat smaller portions. A secondary aim explored the relative contribution of enjoyment of taste and post-meal fullness as determinants of meal satisfaction. Focus groups (N = 42) evaluated consumers’ feelings toward a small reduction in portion size. Thematic analysis of written free association tasks and open-ended group discussions revealed that most participants expected to feel hungry and unsatisfied, which motivated them to consume something else. However, others expected to feel comfortable, healthy, and virtuous. The acceptability of the reduced portion was also determined by meal characteristics (e.g., time and setting) and individual characteristics (e.g., predicted energy requirements). Compared to post-meal fullness, enjoyment of taste was perceived to be the more important determinant of meal satisfaction. In conclusion, interventions should present portion reduction as a marginal modification with little physiological consequence to energy reserves, while emphasizing the positive feelings (e.g., comfort, satisfaction, and self-worth) experienced after consuming a smaller portion. Additionally, focusing on taste enjoyment (rather than fullness) might be a useful strategy to maintain meal satisfaction despite a reduction in meal size.

## 1. Introduction

Individuals with overweight or obesity may be faced with profound negative health consequences, which, in turn, can have a huge economic impact on both an individual and societal level [1]. Being overweight or obese is largely preventable [2]. However, it is widely acknowledged that our modern food environment promotes overconsumption [3,4,5]. Food portion sizes have increased drastically in the United States since the 1970s [6] and, to a lesser degree, in the United Kingdom since the 1980s [7]. Therefore, several researchers have suggested that larger portions are a contributing factor to increasing rates of obesity [6,8,9]. Consistent with this hypothesis, individuals consume more food and non-alcoholic beverages when served larger portions, than when served smaller portions [10]. Importantly, this ‘portion size effect’ appears to be sustained when participants are exposed to larger portions for several days [11] or weeks [12]. Therefore, several researchers have advocated that portion size interventions are required [10,13,14,15].

A recent economic analysis of the impact and cost-effectiveness of 44 obesity interventions in the United Kingdom predicted that reductions in portion size (even as low as 1% to 5%) would have the greatest impact in reducing obesity prevalence [16]. In March 2018, Public Health England proposed a calorie reduction program, which challenges the food industry to achieve a 20% reduction in calories in certain products by 2024. The report suggests that the 20% reduction in calories can be achieved by reforming product recipes (e.g., reducing energy density of food and drink) and/or reduction of portion size [17].

Experimental evidence suggests that consuming smaller portions might reduce overall energy intake (kcal) because appetite control does not closely track changes in energy balance [18] and people only partially compensate for smaller portions [5,19,20]. For example, Rolls and colleagues found that participants who were served meals and snacks that were 25% reduced in portion size ate less across two days by 231 kcal per day but reported similar levels of hunger and fullness [19]. Furthermore, research suggests that serving people smaller portions might also recalibrate what is perceived as a normal portion size and encourage them to choose smaller portions in the future [21].

However, the success of portion size interventions relies heavily on consumer acceptability and, to date, research on this topic is limited [15,22,23,24,25,26,27,28]. Identifying potential barriers to reducing portion size is vital to ensure the success of any intervention for two reasons. First, the intervention will be undermined if consumers experience decreased meal satisfaction and, as a result, engage in compensatory behavior such as consuming two of the smaller portions or purchasing supplementary items. Second, reducing portion size requires cooperation with the food industry, which is unlikely unless the intervention is acceptable to consumers (otherwise, products will be rejected). Two studies by Vermeer and colleagues in the Netherlands assessed consumer and manufacturer acceptability of point-of-purchase interventions aimed at reducing portion size [22,23]. Their results suggest that, while consumers agree with reducing the portion size of unhealthy energy dense foods (e.g., pizza and candy bars), they were hesitant about a general reduction of portion size in supermarkets and restaurants. Consumers believed that manufacturers should not be able to control how much they consume and were concerned that the reduced portion size would not meet their energy requirements [23]. A limitation of these studies is that the magnitude of portion reduction was not specified. Therefore, participants could have considered a range of portion sizes (e.g., a quarter, half, or three-quarters of a portion) when reflecting on their opinions of portion reduction. Since Public Health England is suggesting a small reduction in portion size [17], and experimental evidence suggests that this magnitude of reduction would be effective in reducing total energy intake, it is vital to assess consumer acceptability of a small reduction in portion size [19,20,21,29]. Vermote and colleagues interviewed a small proportion of Belgium consumers after they had taken part in a portion-size field intervention where portions of French fries were reduced by 20% over a four-day period [24]. The portion size intervention was effective in reducing intake of French fries with little difference in level of satiety after eating. However, the majority of participants noticed the reduction in portion size and disagreed that the smaller portion of French fries should be implemented permanently. For the first time, the current study utilized qualitative methods to evaluate systematically consumers’ feelings towards a small, specified reduction in portion size at a specific meal (lunch).

A recent study asked participants to taste and rate their anticipated satisfaction, eating enjoyment (enjoyment of taste), and fullness for several meals and snacks in varying portion sizes [30]. Anticipated eating enjoyment and anticipated fullness from the meal contributed independently and substantially to anticipated meal satisfaction. Furthermore, anticipated meal satisfaction for half portions was not much lower than for full portions of the same meals, which suggests that there is scope to reduce portion size while substantially maintaining meal satisfaction. These findings support the theory that, in humans, ‘meal satisfaction’ is thought to reflect a combination of the reward experienced during the meal (enjoyment of taste) and the fullness that is experienced at the end of the meal [18]. Therefore, a secondary aim of the current study was to use qualitative methods to explore the relative contribution of enjoyment of taste and post-meal fullness as determinants of meal satisfaction in our sample of consumers.

## 2. Materials and Methods

### 2.1. Design

We chose to use focus groups and included men and women of various ages to encourage a diverse set of perspectives [31]. Participants were shown images of a familiar food and were asked to choose the portion they would ideally consume for lunch. They were then shown an image that was 25% smaller and were asked to imagine consuming this portion instead. A 25% reduction in portion size was selected based on the study of Rolls and colleagues [19]. An advantage of our approach was that a visual demonstration of the reduction in portion size was included before participants were asked to reflect on how they would feel if they were to eat this smaller portion for lunch. Participants were told that the new portion was 25% smaller than their original portion. This reference to percentage reduction was specified to ensure the manipulation was standardized across participants.

### 2.2. Participants

In an attempt to recruit a representative sample of the local population, an advertisement was posted in a local newspaper (the Bristol Post) in Bristol, United Kingdom. It was recommended that focus groups remain small (e.g., 6–12 members) and that each group is relatively homogenous [31]. To ensure that the opinions of participants with different genders and of different ages were equally represented, the focus groups were stratified by gender (male and female) and age (18–35 years, 36–55 years, and 56+ years). Accordingly, we recruited 42 participants to take part in six focus groups: females 18–35 years (*n* = 6, BMI, *M* = 22.3, *SD* = 1.4), females 36–55 years (*n* = 7, BMI, *M* = 24.5, *SD* = 5.9), females 56+ years (*n* = 7, BMI, *M* = 27.9, *SD* = 4.3), males 18–35 years (*n* = 8, BMI, *M* = 23.3, *SD* = 2.7), males 36–55 years (*n* = 8, BMI, *M* = 25.3, *SD* = 2.9), and males 56+ years (*n* = 8, BMI, *M* = 27.8, *SD* = 6.4). Ethical approval was granted by the University of Bristol’s Faculty of Science Human Research Ethics Committee (REF: 23561). Additional focus groups would be conducted if theoretical saturation was not achieved following the analysis of the first six groups [32].

### 2.3. Procedures

Each participant who signed up for the focus groups was provided with an information sheet that instructed them to abstain from eating and from drinking beverages (excluding water and coffee/tea without milk) for a period of three hours prior to the session so that they would arrive moderately hungry. Each focus group was conducted from 12:00 pm to 2:00 pm on weekdays. Upon arrival, every participant reviewed the information sheet about the focus group and provided written consent. Every participant was then provided with four different picture booklets. Each booklet contained pictures of one of the four commercially available foods: Spaghetti Bolognese (Sainsbury’s, Bristol, UK), Cheese Sandwich (Sainsbury’s, Bristol, UK), Vegetable Biryani (Sainsbury’s, Bristol, UK), or Pepperoni Pizza (Asda Stores Ltd., Bristol, UK). Energy density and macronutrient content for each food can be found in Table 1. Participants were asked to choose the booklet that contained the meal that they ate regularly for lunch or with which they were most familiar. Each booklet contained 50 pages. On each page was an image of a portion of food in 20-kcal equicaloric steps ranging from 20 kcal (page 1) to 1000 kcal (page 50). Particular care was taken to ensure that each food was photographed on the same white 255-mm diameter plate and to maintain constant lighting conditions and a viewing angle in each image.

Once all participants had chosen their booklet, the moderator provided each participant with a white 255-mm diameter plate that was identical to the plate displayed in the portion size images. Participants were told to use the plate as a size reference when viewing the photos in the booklet. Participants were instructed to look through the booklet and to imagine that they were going to eat that food for lunch right now. The moderator emphasized that participants should imagine that no other foods would be available. Participants were asked to choose the page that displayed the portion that was closest to the portion that he or she would choose to consume for lunch. After participants had chosen their ‘ideal portion,’ the moderator asked participants to use a pink sticky note to mark the page. The moderator then used a look-up table to instruct participants to flip to a second page, which represented a 25% reduction of the original portion chosen. Participants were instructed to mark the second page with a yellow sticky note to facilitate flipping between the two portion sizes during discussions. Participants were encouraged to compare the two portion sizes. The moderator instructed the participants to imagine that, instead of eating what might be their perfect amount of food right now (the portion marked with a pink sticky note), they now had 25% less (the portion marked with the yellow sticky note).

Since we were interested in exploring two specific questions (feelings toward reduced portion size and the relative importance of enjoyment of taste and post-meal fullness in determining meal satisfaction), semi-structured interviews were utilized. Using semi-structured interviews allowed for flexibility in the discussions in case participants generated ideas that the researchers had not preempted. Before engaging in the group discussions, participants were asked to complete a written free association task [26] for five minutes to organize their thoughts about how eating the smaller portion for lunch would make them feel. Using ideas generated from the free association task to direct the subsequent discussion allowed for the agenda to be participant-driven and unabridged [33]. Participants were asked to write freely in response to the question “How would eating the smaller portion size make you feel?” After the written free association task, the moderator prompted a group discussion on this topic. To ensure that the discussions were participant-led, the group was given freedom to discuss the general questions posed by the moderator. The moderator intervened only when clarification was required, or the discussions digressed from the research questions. After the group discussion, participants were instructed to complete a second written free association task where they responded to the question “Any additional thoughts or changes of opinion?”

To address the secondary aim of the study (exploring determinants of meal satisfaction), the moderator proposed the question “So, on balance, do you feel that meal satisfaction is more about enjoyment of taste or more about feeling full?” Participants were invited to have a group discussion on this topic. After the group discussion, the moderator measured and recorded each participant’s height and weight in a private room. Participants were then debriefed and were reimbursed with £15 in appreciation of their time.

### 2.4. Analysis

Since the first free association task was a tool to generate ideas for discussion, the results presented in this study are predominantly from the group discussions. However, despite the moderator’s attempt to ensure that all participants contributed equally to the discussion, some of the ideas generated during the first free association tasks were not mentioned during the group discussions. Yet, responses during the second free association task only reiterated ideas generated during the discussion. Therefore, the analysis combined data from both the first free association task and the group discussions.

Discussions during each focus group were audio-recorded and later transcribed verbatim onto a computer by the moderator. The written data from the free association task was also typed up. The free association task data and the transcriptions did not identify individual participants since the researchers were only interested in the group consensus. The free association task data and the transcriptions were analyzed using NVivo version 10 (QSR International UK Ltd., Warrington, UK, 2012). Thematic analysis, with a theoretical approach, was used to answer the following questions: (1) “How do consumers feel about a reduced portion size?” and (2) “Is meal satisfaction more about the enjoyment of taste or feeling full?” A primary rater developed a coding framework. The coding framework began with the categorization of negative, neutral, and positive responses to the above questions. The primary rater then coded free association task responses and open-ended discussions for all six focus groups based on the coding framework. When coding led to the emergence of themes, the primary rater refined the contents within each of those categories by periodically discussing and verifying the development of themes with a secondary rater. Themes that did not have enough supporting data or did not directly answer our research questions were discarded. After the primary rater completed the analysis, the secondary rater coded a proportion of the interviews to assess interrater reliability. To assess interrater reliability, it is recommended that at least 10% of the transcripts be coded by a second rater [34]. The free association task from one focus group and the open-ended discussions from another focus group were randomly selected (16%) and coded by a second rater. Interrater reliability (Cohen’s kappa coefficient) was found to be 87% or higher (values of 81%–100% are regarded as indicating almost perfect agreement [35]). Since both coders agreed that theoretical saturation had been achieved, additional focus groups were not conducted. Lastly, themes discussed by both genders and more than one age group were identified as the core themes, which are described below.

## 3. Results

### 3.1. Attitudes and Feelings about Eating a 25% Smaller Portion for Lunch

Many participants believed that eating the smaller portion would be insufficient, which would leave them feeling hungry, unsatisfied, and wanting to eat more.
“Unsatisfied and disappointed by the portion size. Still hungry after eating, would not be receiving the calories required.”*(Males, Aged 18–35)*
“Hungry! It’s nowhere near enough. I would need to have something else.”*(Females, Aged 18–35)*
“Would make me feel hungry still. As a meal would not be enough for my needs. I will have to eat something extra.”*(Females, Aged 36–55)*
“I would still be hungry and would feel that I wanted something else to eat.”*(Males, Aged 36–55)*

Some participants were more extreme in their reactions. References were made to deprivation and starvation. Others expressed a preference to forgo the meal completely as the effort required to prepare and clean up the meal would not be worth the size of the meal.
“Starved…the portion size that’s selected, to me, looks more like a sort of an appetizer, and I would want the same again.”*(Males, Aged 56+)*
“So now I feel really deprived... I’m not justifying it at all. I am just concerned about being gratified.”*(Females, Aged 56+)*
“The portion may not even be worth eating at all, since it would leave me craving more. I’m not on a diet so it would definitely not satisfy me.”*(Males, Aged 18–35)*
“Looks quite pathetic on the plate. Barely worth getting the plate and cutlery dirty for. Would probably still feel hungry afterwards/no significant decrease in hunger.”*(Males, Aged 18–35)*

Participants also discussed how the reduced portion size would affect their mood. Specifically, they anticipated feeling annoyed (usually regarding value for their money). “I definitely would not be able to save any to have for lunch the next day, which I sometimes do to save money, which left me feeling marginally annoyed” *(Males, Aged 36–55)*. They also imagined that they would feel disappointed, “Sad—like [they] missed out,” *(Males, Aged 18–35)*, or had been misled, or even robbed.
“A bit cheated!”*(Females, Aged 36–55)*
“I was not given what I asked for.”*(Females, Aged 56+)*
“A bit like I have been robbed.”*(Males, Aged 18–35)*
“It is like it has been tampered with.”*(Males, Aged 18–35)*
“Well, I actually feel short-changed because of expectations.”*(Males, Aged 56+)*

Participants agreed that the reduced portion size at lunch would not keep them full until dinner so they felt that it was inevitable that they would need to consume something else before, during, or after the meal (like a snack, side dish, dessert, or a beverage) to make up for the “missing” food from the meal. There were references made to both low-calorie and high-calorie additions to compensate for the reduction in portion size.
“I would be happy to eat this, but would buy something like a packet of crisps or a chocolate bar to supplement it… to make me feel full.”*(Females, Aged 36–55)*
“I’ll feel hungry sooner than usual in the afternoon. I might eat more supper to compensate for that.”*(Males, Aged 18–35)*
“I would need a piece of cheese or fruit after to feel satisfied though.”*(Females, Aged 18–35)*
“May need some snacks in an hour or so, or a drink with this meal may be more satisfactory.”*(Males, Aged 18–35)*
“I would add a lot of salad ingredients as this would make the plate fuller and, as I would have eaten more, I would feel more satisfied.”*(Females, Aged 56+)*
“The idea of a drink with food is interesting. I would normally have juice or a soft drink with this meal. Adding this may change my opinion about feeling dissatisfied after eating this meal to feeling fully satisfied.”*(Males, Aged 18–35)*
“I was definitely unsatisfied by the smaller portion, but was still wondering about a snack, pudding, is supper going to be earlier, all those types of things.”*(Males, Aged 36–55)*

However, participants recognized that there were certain situations in which a smaller portion size would be acceptable. The most agreed upon example was if the individual was trying to lose weight, “if I wanted to lose weight…I think that would be a perfectly acceptable size” *(Males, Aged 18–35)*. One participant explained that having a different mindset would allow the reduced portion size to be satisfying, “if I had to think that I had to stay healthy and I was on a diet then I think I would have to adjust my thinking” *(Females, Aged 36–55)*. Other examples of when a reduced portion size was acceptable depended on characteristics of the meal, setting of the meal, and an individual’s lifestyle (Table 2 contains further details). Many also pointed out that the type of food was an important factor. For example, one participant explained that “[the] 25% reduction…was not dramatic for [him] because of the type of food” *(Males, Aged 36–55).* Whereas reductions in a plate of pasta or rice were rather unnoticeable. Three quarters of a sandwich or a pizza seemed incomplete and strange. For example, participants made statements such as, “You can see with your eyes that it has been cut” *(Females, Aged 56+)*, “It looked like someone had taken a bite from it” *(Females, Aged 36–55)*, “It looks like it has been tampered with” *(Males, Aged 18–35)*, and “I feel like if we are eating this kind of portion of a sandwich size, I feel like I was robbed…you normally would have two slices of bread, you would make a sandwich, you eat it, but there is a corner missing” *(Males, Aged 18–35)*. Themes generated during the second free association task (additional thoughts or changes of opinion) mirrored those themes covered during the discussion. The only exception was that “eating more slowly” gained popularity as a potential strategy to consume a smaller portion while maintaining meal satisfaction.

Despite the abundance of negative sentiments, some participants did recognize that, although they may originally feel disappointed by the reduced portion size, after eating the meal (especially if they ate slowly), they would be content. It was mentioned that enjoyment of the meal would be reduced, but that overall, they would feel satisfied (although not necessarily equally satisfied). Some noted that they might even feel as though the bigger portion size would have been unnecessary or too much and that eating the smaller portion would leave them feeling more comfortable, healthier, and virtuous afterwards.
“I have not eaten anything since 9 in the morning, so my first reaction was to choose the maximum portion size… but now if I think that, like, if I am given even this portion size then it will leave me equally satisfied, cause that is your first reaction when you are hungry.”*(Females, Aged 18–35)*
“It would make me feel like eating it slower, so psychologically I would feel as full.”*(Females, Aged 56+)*
“I am always tempted if I am hungry to eat until I am almost over-full sometimes… whereas…if I eat less, then I feel less bloated and probably just as satisfied. But …maybe later on in the day I will probably will start to be hungry sooner, but I think it is probably nicer sometimes to eat smaller meals more often, smaller portion size more often.”*(Females, Aged 18–35)*
“At my age…I do not need…vast amounts of food, so I’m quite happy…if then tomorrow I’m beginning to feel a bit peckish, well I might have a bit more. But …it would not bother me.”*(Males, Aged 56+)*
“I would feel it was a bit paltry actually…you might feel, like…I am not sure that will fill me up, but I think after I had eaten it, I probably would think it had filled me anyway. I think it probably would. And I would probably feel quite good with myself for having had a small portion.”*(Females, Aged 36–55)*
“Small portions…there are reasons why it might be smaller and it is kind of OK, and actually you realize that you could probably eat less.”*(Females, Aged 36–55)*
“I would be OK with eating this portion if I had to. I am not starving and could easily put up with it. But the feeling of pleasure (or gluttony, I suppose) would be slightly reduced. I might eat more slowly to get full enjoyment out of the reduced amount of food.”*(Males, Aged 56+)*
“I would be satisfied…with the smaller portion and, therefore, …no longer hungry, in all likelihood, but probably still looking for some sort of snack or something as a cake, packet of crisps, something like that.”*(Males, Aged 36–55)*
“On reflection, I think I would probably be okay with this portion size. Someone mentioned that if they had a smaller portion they would eat more slowly and actually I think if I did this, I would get fuller quicker and, therefore, be able to eat less.”*(Females, Aged 18–35)*

### 3.2. Is Meal Satisfaction More about the Enjoyment of Taste or Feeling Full?

An overwhelming majority prioritized enjoyment of taste over feeling full from a meal.
“I would just say ‘taste’ cause … I like things to taste nice. I do not like blandness, so even if I was not that full, if I had something tasty…I could still feel quite satisfied…cause for me, eating is more of…it is nice to feel full, I do eat a lot, but…it is more than one experience. If I had to just choose the one, I would say ‘taste’ for me.”.*(Females, Aged 36–55)*
“If someone said to me, ‘You know, this is tasteless, but it would fill you up’, I’d say, ‘Forget it, I’ll go hungry.”*(Females, Aged 56+)*
“I think enjoyment, because there is no point in eating it, you know, if you are not going to enjoy it, and I suppose, if you really enjoy something, you would rather have a smaller portion of it.”*(Females, Aged 56+)*
“If you were to ask me…one or the other, is it taste or fullness? I would say ‘taste’.”*(Males, Aged 36–55)*

However, many participants abstained from responding, stating that it was too difficult to choose one or the other. They emphasized that the decision was not a simple binary choice but involved a “combination of factors.” Although the factors that did influence this decision were very specific to the individuals (and, therefore, only listed by one or two of the focus groups), Table 3 lists some examples.

## 4. Discussion

The aim of this study was to explore consumers’ attitudes and feelings toward a small reduction in portion size at lunch and to understand the determinants of ‘meal satisfaction’ (enjoyment of taste or post-meal fullness). Although it is widely advocated that reducing portion size might be a useful tactic for weight maintenance and/or weight loss [16,17,19,21], potential barriers to change the current standard portion sizes need to be minimized, or this strategy will be ineffective. Overall, there was a tone of reluctance among the groups. The majority were adamant that they would be dissatisfied with a small (25%) reduction in portion size. Reasons for this were: (i) effects on hunger both at the end of the meal and during the inter-meal interval, (ii) feeling the need to ‘compensate’ by eating other foods or eating more often, (iii) negative effects on mood, and (iv) feeling that the product had been ‘tampered with.’ These negative effects of portion size reduction will be discussed in turn below. On the other hand, several individuals acknowledged that there were some positives associated with consuming a smaller portion. Specifically, that they would feel more comfortable and virtuous. However, we note that many of the participants who discussed positives associated with portion reduction also added qualifying statements such as, “If wanted to lose weight…I think that would be a perfectly acceptable size,” “I may need some snacks in an hour or so, or a drink with this meal may be more satisfactory,” and “I would say that eating the smaller portion size would leave me satisfied but perhaps craving more, although I know it would be better to eat less.”

### 4.1. Negative Views toward a 25% Smaller Portion at Lunch

The most common criticism for why a smaller portion for lunch would not be suitable was that individuals would not be satiated immediately after the meal or that they would be hungry before their next meal. Consistent with the findings of Vermeer and colleagues [23], participants were concerned that the reduced portion size would not meet all energy requirements, especially for active individuals. Furthermore, one participant contrasted physical and mental exertion (‘decorating your home’ versus ‘concentrating at University’) and felt that the latter would require a greater intake of food. However, these expectations that energy requirements will not be met have little physiological basis. Specifically, the portion reduction used in this study was a trivial amount compared to total body energy reserves [5] and the reductions were specific to the individual (i.e., 25% reduction from their ideal or usual portion size). Furthermore, some participants expressed such contempt of the reduced portion size that they stated that they would prefer to eat nothing at all, which suggests that their concern was less about energy requirements and more about deriving pleasure from the eating experience.

Participants also believed that the smaller portion size would cause them to compensate for the ‘missing food’ by adding a side dish or beverage to the meal, having dessert immediately after the meal, having a snack during the inter-meal interval, or eating more at their next meal. Therefore, a potential barrier to portion reduction is not only that consumers might purchase several smaller portions of the same food [23], but that they might supplement with other foods (or beverages) during or after the meal. However, these consumer expectations are inconsistent with the experimental evidence. For example, Rolls and colleagues [19] found that 25% reductions in portion size over two days led to decreased energy intake. These effects were sustained from meal to meal and there were no significant differences in self-reported hunger and fullness ratings. Furthermore, other researchers have employed larger reductions in portion size and still did not observe energy compensation at subsequent meals [20]. Similarly, while consumers in this study expected that they would choose to have “crisps” or a “soft drink” if they were eating a smaller portion, research has demonstrated that intake of high energy dense snacks and beverages (compared to lower energy dense alternatives) is increased when the portion size of the main meal is increased [36].

One explanation for the inconsistency between consumers’ expectations and the experimental evidence is that consumers’ expectations are, in fact, just that—expectations—and that their actual experiences would match the experimental evidence if they were willing to try consuming the reduced portion size. However, participants from the focus groups referred to compensating for the reduced portion not only in terms of how much they would eat, but also in terms of how often they would eat. Experimental studies, which mostly utilize pre-load test-meal designs, typically do not necessarily pick up on the latter, which highlights another advantage of using qualitative methodology. Therefore, future research should explore how reducing portion size affects compensatory behavior in terms of the types of snacks and beverages that are chosen with a meal and in terms of the frequency of eating and drinking events. For example, a recent study found that participants were more likely to select a higher-energy beverage to drink with their meal if the food portion size was small [37].

Many of the comments about how smaller portion sizes would affect mood revolved around monetary value. Although some participants did mention feeling sad or disappointed, a more common expectation was that they would feel annoyed because they would not be receiving their money’s worth or would not be receiving what they expected to get. These findings are also consistent with the Vermeer and colleagues’ study [23], in which participants disliked the idea of manufacturers controlling the portion sizes on their behalf. A recent study that investigated the idea that the portion size effect is influenced by value for money found that energy intake from large portions was not affected by the cost of the meal [38]. While additional research is needed, it is possible that the decreased value for money associated with smaller portion sizes is less influential than expected [39]. Results from another recent study suggest that the inherent value of food may be more pertinent instead. The portion size effect was diminished when participants were given the option to save leftover food for later (thus, reducing food waste) [40].

It is also important to note that many of these comments came from participants who chose the sandwich or pizza option for the exercise. Therefore, these feelings may be specific to foods that appear ‘tampered with’ when using an equicaloric reduction. For example, participants stated, “It does not look as appealing as [the original portion chosen]…looks like someone has taken a bite from it,” and “I chose my ideal portion size because the pieces of bread were complete. Here, they are not, so it is noticeable that some is missing.” Future research should investigate whether a legitimate difference exists in how much homogenous (e.g., pasta) versus discrete unit (e.g., sandwich) foods can be reduced. If consumers are going to be more reluctant to purchase reduced portions of discrete unit foods compared to homogenous foods, food manufacturers might want to reduce portion size using a methodology that is less noticeable (e.g., serving ‘complete’ pizzas with smaller diameters).

### 4.2. Positive Views toward a 25% Smaller Portion at Lunch

In contrast with the findings of Vermeer and colleagues [23], opinions expressed during the focus groups were not entirely negative. Although many of the participants met the suggestion to reduce portion size with resistance, it is possible that they may have been content with their body weight and, therefore, the reduction was not appropriate for them. For example, many agreed that a smaller portion size would be acceptable for an individual trying to lose weight. Similarly, Vermeer and colleagues [23] note that “many [of their] participants supported portion-size interventions not because they found them as personally relevant but because they thought they would help combat obesity...”. For lean participants in our study, the reduction may have been difficult to seriously consider, which leads to feelings of annoyance. Therefore, focus groups conducted with individuals who are discontent with their body weight or actively trying to lose weight might demonstrate a larger proportion of positive findings.

Nonetheless, portion size interventions are useful not only for losing weight, but also for maintaining a healthy weight. Participants discussed additional factors that contributed to the acceptability of small portion sizes (and, therefore, would be applicable regardless of the consumer’s weight status). Acceptability of a small portion size depends on which meal is being considered. In this study, we proposed that participants imagine reducing their lunch portion by 25%. The proposition that breakfast or dinner instead be reduced by 25% might be less or more acceptable to consumers, and there might be individual differences. For example, individuals differ in whether breakfast, lunch, or dinner is their largest meal. Where and with whom the meal is eaten also needs to be considered. Special occasions (eating at a restaurant or at a friend’s house) may be situations in which reduced portion sizes might be less appropriate, as celebratory events are strongly associated with feasting [41,42].

The length of time between meals is also a consideration. A small portion size would be more acceptable if the inter-meal interval was shorter. A virtual study, which asked participants to choose a portion size to have at lunch revealed that participants who were told to imagine they would be having dinner at 9:00 pm chose larger portion sizes than those participants who were told they would have dinner at 5:00 pm [43]. If individuals reduce the size of their meals, but, subsequently, eat more frequently, as one participant stated, “it is probably nicer sometimes to eat smaller meals more often, smaller portion size[s] more often.” They might not reduce their overall intake. Therefore, consumers need to understand the difference between productive and counterproductive compensatory techniques. One example mentioned by participants was “eating slowly.” Participants seemed to like the technique of “eating slowly” because it would work to increase both enjoyment during the meal and post-meal fullness. Eating slowly might increase (1) enjoyment during the meal because it allows the consumer to enjoy salient features of foods and the environment [44] and (2) post-meal fullness by allowing more time to feel the effects of postprandial satiety hormones [45].

Some participants admitted that there would not be profound differences on appetite and feelings of satisfaction between consuming the original portion size and the reduced portion size. This finding is consistent with the results of Vermote and colleagues’ study [24] in which the majority of participants (68%) reported that the smaller portions were sufficient. However, only 32% supported permanent implementation of the reduced portion sizes. Similarly, in the present study, several participants who agreed that the reduced portion size would be sufficient still expressed concern that they would need to eat additional food soon after to sustain those feelings of satisfaction or to stave off hunger. There were also expectations of feeling disappointed at first sight of the meal or having a less pleasurable experience while eating. However, participants believed that these adverse effects could be outweighed by the benefits of the reduced portion size (relieving their hunger while leaving them comfortably full and reducing feelings of eating-related regret and increasing feelings of self-worth). In terms of comfort, while selected portion size is predicted by expected satiety [46], there may be a threshold at which satiety is no longer preferable. With large portions, expected satiety becomes a negative predictor of selected portion size [46] and research with rodents demonstrate that “excessive” satiety can be aversive [47]. In terms of self-worth, it has been suggested that self-worth enhancement can help overweight individuals become more self-reliant and increase the efficacy of weight loss treatments [48]. Nonetheless, participants from the focus groups explained that, despite recurring experiences of regret after overeating, when hungry, they have a strong temptation to eat a great deal (sometimes to the point of excessive satiety). It is not until the end of the meal, once hunger is low, that they recognize that they had eaten too much and that a smaller portion would have been more suitable. A potential mechanism for why individuals might be more content after consuming a smaller portion is due to increased sensory-specific satiety, which limits intake and avoids excessive satiety. While more research is needed, it has been suggested that smaller portion sizes induce smaller bite sizes, which increases oral sensory stimulation and sensory-specific satiety [49,50].

### 4.3. Meal Enjoyment

Initially, participants appeared to be more concerned by fullness. The most common complaint for why a smaller portion size would not be suitable was lack of satiation at the end of the meal, or that satiety would not last long after the meal. Fewer participants referred to enjoyment during the meal (feeling deprived, disappointed, or like the meal was over too quickly). However, when explicitly asked if they would prefer to feel full or enjoy the taste of the meal, the majority opted for enjoyment of taste. It should be pointed out this was not a clear-cut decision for most, as many participants commented that it was impossible to decide between taste and fullness or that their decision was dependent on the situation. These results complement recent experimental findings that, while both eating enjoyment (enjoyment of taste) and fullness from the meal predict meal satisfaction, eating enjoyment is the stronger predictor [30]. The results taken together suggest that focusing on enjoyment of taste (rather than fullness) might be more effective in maintaining meal enjoyment when marginally reducing portion size.

## 5. Conclusions

The results of this study demonstrate that ensuring consumer compliance to portion reduction will be a daunting task. Most participants were reluctant to accept portion reduction, and those who were open to the modification did not appear entirely convinced. It is possible that the overt manipulation used in this study (highlighting the degree of portion reduction) amplified the prevalence of negative reactions. However, previous research found that individuals readily discriminate between differences in portion sizes [51], which suggests that the participants in the present study would have been aware of the reduction even if they were not explicitly told that the second portion was reduced by 25%. In addition, highlighting the degree of reduction reflects actual packaging of manufactured food (e.g., 30% less sugar) [52]. This practice is likely to become increasingly prevalent in the future due to the calorie reduction program recommended by Public Health England [17]. While some manufacturers may attempt to disguise portion reductions, consumers’ ability to discriminate between different portion sizes suggest that any portion interventions are unlikely to be covert [51]. One strategy to reduce resistance to portion reduction might be to tackle consumers’ misunderstanding of physiology. For example, putting portion reduction into perspective for consumers can be done by comparing it to the original portion or total body energy reserves and emphasizing the lack of compensation seen in experimental studies. This, in turn, should help manage their expectations of consuming marginally smaller portion sizes.

It is promising that many participants shared previous experiences of feeling satisfied by small portion sizes and recognizing that the actual experience did not match their prior expectations. However, despite that knowledge, many remained unenthusiastic about portion reduction. Therefore, simply providing consumers with scientific facts might be inadequate. One possible strategy to tackle consumers’ unwillingness to change is to use framing techniques [42]. While it might seem somewhat patronizing, consumers may benefit from reminders that we often feel excessively satiated from a portion that originally seemed ideal when hungry, and highlight the alternative of feeling comfortably satiated, yet satisfied. Consumers should be reminded that feeling guilty after eating too much or feeling deprived after eating too little are not the only two possibilities, but that eating somewhat less might result in feeling both satisfied and virtuous.

Meal satisfaction is a complex interaction between enjoyment of taste and post-meal fullness, which suggests that strategies for portion reduction need to focus on both postprandial fullness and enjoyment of taste [17]. However, if maintaining both enjoyment of taste and postprandial fullness is impractical, our results suggest that the former should be emphasized. Strategies that focus solely on developing satiety-enhancing ingredients for food and beverage products might have limited success. The suggestions for how to increase consumer acceptability of reduced portion sizes, as outlined in this study, are particularly timely since recent government recommendations in the United Kingdom strongly advocate for portion reduction in the food industry [17].

## Figures and Tables

**Table 1 nutrients-11-01054-t001:** Energy density and macronutrient content for 100 g portions of each of the three foods.

Food (100 g)	Energy (kJ)	Energy (kcal)	Protein (g)	Fat (g)	Carbohydrates (g)
Spaghetti Bolognese	593	141	7.2	5.3	16.2
Cheese Sandwich	1382	330	19.6	22.9	15.3
Vegetable Biryani	657	127	2.9	4.8	16.1

**Table 2 nutrients-11-01054-t002:** The feasibility of feeling satisfied with a 25% smaller portion depended on characteristics of the meal and the individual. The themes included below were discussed by both genders and more than one age group.

Characteristics	Quotations
Meal timing	
Time between meals	“My reaction depends on how long it would be until dinner time.” *(Males, Aged 36–55)*
Time of day	“If I was given a smaller evening meal, this would not concern me as much, as I tend to eat very little in the evenings usually.” *(Females, Aged 18–35)* “And at lunchtime, probably lunchtime, that portion might be alright for me.” *(Females, Aged 36–55)*
Meal setting	
In a restaurant vs. at home	“I think it would also depend on whether YOU…prepared that for yourself, or you are in a restaurant. Cause if you have that in a restaurant, you would look at it and think: ‘No, it will not be enough. I am paying all this money’” *(Females, Aged 36–55)*
Alone vs. with company	“If I was eating by myself, it would be different I think... If I was eating with other people…I would be totally happy, with the—with that smaller one…or if I was by myself…doing something at the same time as well, I think mentally engaged with something else…I would be perfectly happy. In fact, if I was with other people and I ate that, and then was offered seconds, I would probably say ‘no’.” *(Males, Aged 56+)*
Individual characteristics	
Activity level	“If you did that in your own kitchen at home, I think it would depend on what you were doing at the time, if you were in the middle of decorating and you just grab yourself something quick, but if you are in the University and you are concentrating all day, you will probably need more.” *(Females, Aged 36–55)*
	“Depending on a person’s lifestyle, the need of food, and the size of food portion will be different. As an example, someone who works in an office, who does not move around much will, in my opinion, need a smaller food portion than someone who has physical work. As individuals, we are all different with different needs.” *(Females, Aged 36–55)*
	“If I knew I was exercising later, like going for a swim or something, then it would probably be a better portion size than before. Because you do not want to have a full stomach before a swim.” *(Males, Aged 18–35)*
Energy needs	“I think different people … they have a different need of… food.” *(Females, Aged 36–55)*
	“Different people will obviously need different amounts of food to make them feel satisfied.” *(Females, Aged 36–55)*

**Table 3 nutrients-11-01054-t003:** Conditions for whether meal satisfaction was more about enjoyment of taste or feeling full after a meal. The themes included below were discussed by both genders and more than one age group.

Conditions	Quotations
Level of hunger	“It also depends on how hungry you are at the time…if you are really, really hungry you take the big bowl, wouldn’t you?” *(Females, Aged 56+)*
Meal type (Breakfast vs. Lunch vs. Dinner)	“With breakfast, you would be more—more towards the filling, and an evening meal more towards taste.” *(Males, Aged 36–55)*
In relation to earlier and later meals	“If you know you have a big meal coming up, you might maybe go for something small but high taste but if you know you are not going to eat for a while, you will go for something bulky but maybe less—less tasty.” *(Males, Aged 36–55)*
Eating alone vs. eating with friends	“If you go out with your friends for dinner, it is more about the enjoyment, obviously. If you are eating a quick lunch on your own, maybe it is not as much about enjoyment.” *(Females, Aged 18–35)*
Eating at home vs. eating in a restaurant	“I think it very much depends if you are going out to a restaurant for a nice meal, you are more focused on how it tastes but if you are just eating your lunch half-way through your day, you are just focused on feeling full, and lasting until the end of the day.” *(Males, Aged 18–35)*
Special occasion	“Yeah, you want it to feel special so you want it to be something new and exciting, not just ‘I want to feel full again’” *(Males, Aged 18–35)* “If you went out for lunch every single day, you would want to feel full, but…if you…went out for lunch once or twice a week, once or twice a month, it would be about taste, it depends how frequently you are having this product.” *(Males, Aged 18–35)*
Expectations	“Indian restaurants usually do huge meals. I went to an Indian [restaurant] once and got a small portion and was horribly disappointed. It tasted great, whereas when you go to other restaurants which you would expect to do smaller portions…more like posh restaurants…you go to them, you expect to get small portions…you are satisfied.” *(Males, Aged 18–35)*

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
