# Peer review of "Identifying Barriers to Reducing Portion Size: A Qualitative Focus Group Study of British Men and Women"

_nutrients, 2019, doi:10.3390/nu11051054_

Round 1
Reviewer 1 Report
This is an interesting project that studies an important topic, however, there is certain information missing from the paper which would make it more relevant for knowledge users and researchers, specifically data on demographics, how responses may have differed between sub-groups and percentage of participants with similar/same themes. Considering that participants were informed of 25% reduction in portion sizes, how are the resulting responses justified?
Specific comments:
Lines 79-86: Suggestion to remove from introduction and place it in methods. Focus the last two paragraphs on stating the gaps and primary/secondary objectives.
Line 100: Any information on the characteristics of individuals who may have had access to said newspaper? Are they in any way different than the local population who may not have access to the newspaper?
What demographic information was collected on the participants?
Line 107: How was the sample size determined?
Lines 111: Clarify whether the participants were provided with the information prior to coming to the session as instructed? How did you ensure that the instructions were followed?
Lines 115-116: Please provide reasoning for why those particular foods were chosen?
Lines 134: Participants were told that the second portion size is 25%, would the results be different if they were not told of this information?
Line 172: What was the coding framework based on?
Line 243 and 287: Table 1 and 2; themes identified seem haphazardly chosen with some characteristics/conditions missing (e.g. preparation, age requirements etc.) is there a justification for how this coding framework was achieved? Demographic factors such as SES and health literacy might impact the results.
Line 291-307: These lines can be condensed as most of them are repeating the results.
Lines 357-358: These lines seem speculative. Is there a study suggesting this?
Line 359-360: Since monetary value is stated as one the reasons for the negative views, are there studies showing how participants react if the monetary value was also in alignment with portion sizes?
Lines 364-365: study would have benefited from collection of this information.
Lines 410: It is unclear what this line is trying to achieve, perhaps elaborate on this further.
Line 421: Study is missing number/percentage of participants who may have agreed on common themes e.g. it is stated that majority opted for enjoyment but what is a majority?
Consider the role that sensory-specific satiety can play in the results (where consumption of larger portion sizes results in less enjoyment) in the discussion (consider research by Herman).
Elaborate on how the results/conclusions facilitate the policy around portion sizes for packaged foods in relation to the intakes/attitudes? The foods chosen do not necessarily indicate packaged foods.
Author Response
GENERAL COMMENTS FROM REVIEWER 1
This is an interesting project that studies an important topic, however, there is certain information missing from the paper which would make it more relevant for knowledge users and researchers, specifically data on demographics, how responses may have differed between sub-groups and percentage of participants with similar/same themes. Considering that participants were informed of 25% reduction in portion sizes, how are the resulting responses justified?
GENERAL RESPONSE TO REVIEWER 1
Thank you to Reviewer 1 for reviewing our manuscript. We have tried our best to address your concerns and respond to your comments. In regards to your general comment, we respond to your comment about the data on demographics and how sub-groups may have differed in their responses below (points #2, #9, and #12). We respond to your question about the impact of informing participants of the 25% reduction on the results below (point #7). We have answered your specific comments below each point, bolded and in purple. Changes to the manuscript have also been bolded and written in the colour purple.
Specific comments:
1) Lines 79-86: Suggestion to remove from introduction and place it in methods. Focus the last two paragraphs on stating the gaps and primary/secondary objectives.
Thanks for this suggestion. We have moved the text to the methods section “2.1 Methodology”, lines 93-99.
2) Line 100: Any information on the characteristics of individuals who may have had access to said newspaper? Are they in any way different than the local population who may not have access to the newspaper?
The newspaper was the Bristol Post (regional daily newspaper) which has 9 million monthly subscribers. It is possible that this recruitment method limited participation to individuals from mid to high socio-economic status groups. We did not collect information on SES. The newspaper is now mentioned by name in the manuscript. We have also changed the text “in order to recruit a representative sample” to “in an attempt to recruit a representative sample”. These changes have been made to be more transparent about the population we sampled from.
3) What demographic information was collected on the participants?
Lines 108-112 report all of the collected information which was age, sex, BMI.
4) Line 107: How was the sample size determined?
We followed established guidelines for determining sample size when conducting focus groups:
Lines 105-112: It is recommended that focus groups are small (e.g., 6-12 members) and that each group is relatively homogenous [25]. To ensure that the opinions of participants with different genders and of different ages were equally represented, the focus groups were stratified by gender (male and female) and age (18-35 years, 36-55 years, and 56+ years). Accordingly, we recruited 42 participants to take part in six focus groups: females 18-35 years (n = 6, BMI, M = 22.3, SD = 1.4); females 36-55 years (n = 7, BMI, M = 24.5, SD = 5.9); females 56+ (n = 7, BMI, M = 27.9, SD = 4.3); males 18-35 years (n = 8, BMI, M = 23.3, SD = 2.7); males 36-55 years (n = 8, BMI, M = 25.3, SD = 2.9); and males 56+ (n = 8, BMI, M = 27.8, SD = 6.4).
After preliminary analysis, both coders agreed that theoretical saturation had been achieved, and therefore additional focus groups were not conducted. This point was made in the analysis section (line 192-193), but we have also included it at the end of “2.2: Participants” (line 114-115).
5) Lines 111: Clarify whether the participants were provided with the information prior to coming to the session as instructed? How did you ensure that the instructions were followed?
Participants who signed up for the study received an information sheet prior to the focus group session. The information sheet stated the requirement for participants to abstain from food/calorie-containing beverage for 3 hours prior to the focus group session. This information has now been added to the manuscript (line 117-118). Participants were also reminded of this requirement via email the day before their session. At the study sessions, we did not ask participants to confirm that they had abstained. It was not essential that participants were fasted, but we thought it would make the responses more accurate if participants were moderately hungry.
6) Lines 115-116: Please provide reasoning for why those particular foods were chosen?
These foods were selected as they are commercially available well-known foods which have also been used in previous research studies [1–5]. We have now added the description “commercially available” to line 122. We have also included where each food was purchased from on line 122-124.
7) Lines 134: Participants were told that the second portion size is 25%, would the results be different if they were not told of this information?
We agree that the results might have differed if the percentage decrease was not made explicit to participants: participants may have been less cognisant of the decrease in size, and therefore may have shown less resistance towards the reduction. However, research suggests that individuals are easily able to discriminate between differences in portion size [6] so they may have been just as aware of the reduction even without our specific instructions. We did consider both alternatives when designing the study, but we decided to verbalize the percentage decrease to participants as we wanted to standardize the manipulation across participants and use an overt manipulation. This was for a few reasons. Firstly, we wanted to avoid participants having different interpretations of the portion manipulation so that we could systematically assess opinions of portion size. Secondly, we wanted the manipulation to mirror what consumers might see on packaging (e.g., 20% less fat, etc.). Thirdly, we felt it was more important to investigate overt manipulations of portion size as it is unlikely that any portion size intervention will be covert (as mentioned earlier, individuals can discriminate between differences in portion sizes [6]).
More detail discussing the selection of the methodology has been added to lines 100-102. More detail discussing the limitations of specifying 25% reduction to participants has now been added to the conclusion section (lines 457-467).
https://www.bbc.co.uk/news/business-44885840
8) Line 172: What was the coding framework based on?
The text on lines 177-186 have been amended to provide more information on the coding framework.
“Thematic analysis, with a theoretical approach, was used to answer the following questions: (1) “How do consumers feel about a reduced portion size?” and (2) “Is meal satisfaction more about the enjoyment of taste or feeling full?” A primary rater developed a coding framework. The coding framework began with the categorization of negative, neutral, and positive responses to the above questions. The primary rater then coded free association task responses and open-ended discussions for all six focus groups based on the coding framework. When coding led to the emergence of themes, the primary rater refined the contents within each of those categories, periodically discussing and verifying the development of themes with a secondary rater. Themes which did not have enough supporting data or did not directly answer our research questions were discarded.”
9) Line 243 and 287: Table 1 and 2; themes identified seem haphazardly chosen with some characteristics/conditions missing (e.g. preparation, age requirements etc.) is there a justification for how this coding framework was achieved? Demographic factors such as SES and health literacy might impact the results.”
Themes such as those that you suggest (i.e., preparation, age requirement, etc.) are not mentioned in the manuscript because these were not themes which resulted from the coding of the data. The tables list themes which we felt were of less importance to the participants during the discussions (and therefore were not discussed in the main text), but nonetheless need to be reported and considered when interpreting the findings (and thus reported in tables). Table 1 lists themes which were in response to the question, “How would eating the smaller portion size make you feel?”. Table 2 lists themes which were in response to the question, “So, on balance, do you feel that meal satisfaction is more about enjoyment of taste or more about feeling full?” We stated in the text that the emerging themes had to be discussed by both genders and across more than one of the age groups, but we have now included the statement, “The themes included below were discussed by both genders and more than one age group,” in the caption for Tables 1 and 2 to remind the reader of this information.
We agree that factors such as SES and health literacy could influence the results and future research should systematically evaluate their influence. As stated earlier, we attempted to recruit a diverse sample, but we cannot comment specifically on SES etc., as we did not measure these demographics. These topics also did not come up in the focus group discussions.
9) Line 291-307: These lines can be condensed as most of them are repeating the results.
We feel that this summary (although repetitive of the results) is helpful to the reader as they will have just been presented with a large amount of information in the results. In addition, these sentences also lay out the framework for the remainder of the discussion.
10) Lines 357-358: These lines seem speculative. Is there a study suggesting this?
Thank you for pointing this out to us. Lines 374-379 (originally 357-358) are reflecting on one of the results of this study (that participants felt the less homogenous foods appeared “tampered with” when reduced). We have re-worded these lines to make it clearer to the reader that these points are based on the findings of this study.
“Future research should investigate whether a legitimate difference exists in how much homogenous (e.g., pasta) versus discrete unit (e.g., sandwich) foods can be reduced. If consumers are going to be more reluctant to purchase reduced portions of discrete unit foods compared to homogenous foods, food manufacturers might want to reduce portion size using a methodology that is less noticeable (e.g., serving ‘complete’ pizzas with smaller diameters).”
11) Line 359-360: Since monetary value is stated as one the reasons for the negative views, are there studies showing how participants react if the monetary value was also in alignment with portion sizes?
This is an interesting point, and we’ve now referenced a recent study which looked at the role of monetary value in the portion size effect in lines 364-368.
“A recent study which investigated the idea that the portion size effect is influenced by value for money found that energy intake from large portions was not affected by the cost of the meal [32]. While additional research is needed, it is possible that the decreased value for money associated with smaller portion sizes might be less influential than expected.”
In addition, it is unlikely that a 25% decrease in size would be accompanied by a 25% decrease in price, as packaging and production make up the majority of commercial food costs.
12) Lines 364-365: study would have benefited from collection of this information.
In hindsight, we agree that the study would have benefitted from collection of this information, which is why lines 389-391 highlight that the results might differ if distinctions were made between those content and discontent with their body weight. It would be informative to investigate the feasibility of portion reduction in various subgroups (e.g. currently dieting, dietary restraint, SES, etc.,) but in the first instance, our aim was to ask these questions in a broader sense.
13) Lines 410: It is unclear what this line is trying to achieve, perhaps elaborate on this further.
Line 427-430 (originally line 410) refers to the idea that participants recognized the worth of being comfortably full. It points out that while we eat to achieve fullness, there may be a threshold at which satiety is no longer preferable, as excessive satiety can be uncomfortable and averse. Line 410-412 has been edited for clarity:
“In terms of comfort, while selected portion size is predicted by expected satiety [36], there may be a threshold at which satiety is no longer preferable. With large portions, expected satiety becomes a negative predictor of selected portion size [36] and research with rodents demonstrates that “excessive” satiety can be aversive [37].”
14) Line 421: Study is missing number/percentage of participants who may have agreed on common themes e.g. it is stated that majority opted for enjoyment but what is a majority?
As stated in lines 174-176, individual participants were not identified in the transcripts as we were only interested in the group consensus. Therefore, we do not have data on number/percentage of participants who opted taste or fullness. We stated that the majority opted for taste over fullness because when the question “Is meal enjoyment more about enjoyment of taste or fullness” was posed to participants, there were 20 mentions of affirming “enjoyment of taste” in 4 out of the 6 focus groups, but only 3 mentions of affirming “fullness” in 2 out of the 6 focus groups. As we did not record which participants made what statements, we do not know if the 20 mentions were from 5 participants or 20 participants, so we are unable to cite numbers/percentages. However, from the in-depth analysis of the transcripts, we are confident in stating that the majority of participants preferred “enjoyment of taste”.
15) Consider the role that sensory-specific satiety can play in the results (where consumption of larger portion sizes results in less enjoyment) in the discussion (consider research by Herman).
We have added two sentences (line 436-440) considering the role that SSS plays in the results.
16) Elaborate on how the results/conclusions facilitate the policy around portion sizes for packaged foods in relation to the intakes/attitudes? The foods chosen do not necessarily indicate packaged foods.
We would argue that the study foods do indicate packaged foods as one of the reasons for their selection was that they are all commercially available. With the exception of the cheese sandwich, the foods were ready-meals. We have now added the description “commercially available” to line 122. We have also included where each food was purchased from on line 122-124.
Reviewer 2 Report
Lines 39-40, Therefore, several researchers have advocated that portion size interventions are required [9], [12-14].
There is an extra "," between the reference 9 and 12-14. Is there some missing information?
Lines 99, 2.1, Participants
Is there more information collected from study participants beside sex, age, and measured height and weight, such as prior diseases of themselves or family members or occupation?
Because having close experiences with diseases may influence their eating behaviors, such as types and amounts of food consumed. Also, types of occupation (e.g., with nutrition background) may affect their eating behaviors,
Lines 111, Each focus group was conducted from 12:00 pm to 2:00 pm.
Do authors collect information on the day of the week? Eating behavior may differ in weekday vs. weekend.
Lines 115-116...booklets (these contained pictures of either Spaghetti Bolognese, Cheese Sandwiches, Vegetable Biryani, or Pepperoni Pizza).
Can authors add more information as why those 4 food items be selected? Have those food items be tested as "likes" and "dislikes" as each may have preference for the study participants, which ultimately affect the consumption amount and willingness for consumption reduction.
Author Response
Dear Reviewer 2,
Thank you Reviewer 2 for reviewing our manuscript and your helpful feedback. We have tried our best to respond to your comments below each point, bolded and in purple. Changes to the manuscript have also been bolded and written in the colour purple.
Lines 39-40, Therefore, several researchers have advocated that portion size interventions are required [9], [12-14].
There is an extra "," between the reference 9 and 12-14. Is there some missing information?
The comma is there on purpose, it is separating the brackets around “9” and brackets around “12-14”. We did not do this manually, but it was an automatic insertion by the reference software.
Lines 99, 2.1, Participants
Is there more information collected from study participants beside sex, age, and measured height and weight, such as prior diseases of themselves or family members or occupation?
Because having close experiences with diseases may influence their eating behaviors, such as types and amounts of food consumed. Also, types of occupation (e.g., with nutrition background) may affect their eating behaviors,
No, the only data collected on participants was sex, age and BMI. As you point out there are other factors that will influence eating behaviours. This list is very extensive (SES, dietary restraint, diet restrictions, medications, mental and physical illness, ethnicity, culture and so on) and therefore everything could not be included. While we think that the factors you suggest would be very interesting and informative to assess in relation to our research questions, we do not believe that they are necessary to answer our research questions. Finally, we did not keep track of which participants said what, therefore we wouldn’t be able to look at the relationship between individual differences and opinions toward the portion intervention.
Lines 111, Each focus group was conducted from 12:00 pm to 2:00 pm.
Do authors collect information on the day of the week? Eating behaviour may differ in weekday vs. weekend.
All focus groups were conducted during weekdays and we have now added this information to the manuscript (line 120). For pragmatic reasons, we did not conduct focus groups on weekends. We don’t believe that conducting the focus groups during the week versus on the weekend would drastically change the results as participants were asked about their opinions of portion size generally and were asked to reflect on past experiences (which would include eating episodes on both weekdays and weekends).
Lines 115-116...booklets (these contained pictures of either Spaghetti Bolognese, Cheese Sandwiches, Vegetable Biryani, or Pepperoni Pizza).
Can authors add more information as why those 4 food items be selected? Have those food items be tested as "likes" and "dislikes" as each may have preference for the study participants, which ultimately affect the consumption amount and willingness for consumption reduction.
These foods were selected as they are commercially available well-known foods which have also been used in previous research studies [1–5]. We have now added the description “commercially available” to line 122. We have also included where each food was purchased from on line 122-124.
We did not assess liking for the foods but “participants were asked to choose the booklet that contained the meal that they ate regularly for lunch or with which they were most familiar” (as described in lines 125-126).